# Temporal Trends of Acute Hepatitis A in Brazil and Its Regions

**DOI:** 10.3390/v14122737

**Published:** 2022-12-08

**Authors:** Giuliano Grandi, Luis Fernandez Lopez, Marcelo Nascimento Burattini

**Affiliations:** 1Infectious Diseases Division, Hospital São Paulo, Escola Paulista de Medicina, Universidade Federal de São Paulo, São Paulo 04023-062, SP, Brazil; 2Institute of Mathematics and Statistics, The University of São Paulo, São Paulo 05508-090, SP, Brazil; 3Discipline of Medical Informatics and LIM-01 HCFMUSP, Department of Pathology, School of Medicine, The University of São Paulo, São Paulo 01246-093, SP, Brazil; 4Departamento de Medicina Legal, Bioética, Medicina do Trabalho e Medicina Física e de Reabilitação da Faculdade de Medicina da Universidade de São Paulo, São Paulo 05405-000, SP, Brazil; 5Center for Internet Augmented Research and Assessment—CIARA, Florida International University, Miami 33199, FL, USA

**Keywords:** Hepatitis A, Hepatitis A virus, epidemiology, mathematical modelling, trends

## Abstract

Background: Hepatitis A is responsible for 126,000,000 cases of acute viral hepatitis distributed heterogeneously worldwide, with a high disability-adjusted life year (DALY) rate, especially in low-income countries. Data related to Hepatitis A provides information to improve control measures and identify the population at risk. This study aims to analyze temporal trends of Hepatitis A in Brazil and its regions from 2007 to 2018, based on official notification data. Methods: Data related to Hepatitis A reported cases from 2007 to 2018 were fitted to a joinpoint model by Brazilian regions, age groups, and gender, allowing the calculation of average annual percentage change (AAPC) and annual percentage change (APC) to estimate trends of Hepatitis A in Brazil. Findings: From 2007 to 2018, 65,284 Hepatitis A cases notified in Brazil were available for analysis. The Northeast Region reported 18,732 (28.69%) cases, followed by the North Region reporting 18,430 (28.23%), the Southeast Region reporting 14,073 (21.55%), the South Region reporting 7909 (12.11%), and the Central-West Region reporting 6140 (9.4%), respectively. Temporal trend analysis showed that Hepatitis A incidence decreased from 2007 to 2016 in all Brazilian regions for individuals less than 20 years old, but increased in the South and Southeast males between 20 to 39 years after 2016. Conclusions: Hepatitis A endemicity is heterogeneous among Brazilian regions. In addition, an unexpected outbreak of HAV among Southeast and South adult males in 2016 resembles the outbreak in Europe, revealing a vulnerable population that should be prioritized by vaccination programs and control measures.

## 1. Introduction

Acute Hepatitis A is a disease caused by the Hepatitis A Virus (HAV), transmitted mainly through the ingestion of contaminated water and food. Most cases are self-limited and involve lifelong immunity, verified by the persistence of Anti-HAV IgG antibody serum concentrations, as expected [1]. In 2019, this infection was responsible for 159 million cases of acute viral hepatitis and 39,000 deaths due to liver failure [2]. Although deaths related to Hepatitis A correspond to a small fraction of deaths related to viral hepatitis, HAV infections have high disability-adjusted life year (DALY) rates, especially in low-income countries [2,3].

The ratio of individuals with Anti-HAV IgG, i.e., the prevalence of Anti-HAV IgG, is used to measure the Hepatitis A epidemiological status of a world region. Age-related prevalence is used to describe the risk of acquiring Hepatitis A, or the endemicity of Hepatitis A [4]. High-endemic regions, such as South Asia and Sub-Saharan Africa, are marked by poor sanitary conditions, while low-endemic regions are characterized by high-socio-demographic development [5,6].

Hepatitis A endemicity of tropical Latin America is classified as intermediate. However, this world region contains countries with low endemicity, such as Argentina, Uruguay, and Chile, countries with high endemicity, such as Bolivia and Guatemala, and countries with intermediate endemicity, such as Paraguay, Colombia, and Brazil [7].

Brazil is the largest country in population and territory extension of tropical Latin America and is composed of five distinct socio-demographic regions—North (seven states), Northeast (nine states), Southeast (four states), South (three states) and Central-West (three states and the Federal District) [8].

Although Brazil is classified as low-intermediate endemic for Hepatitis A, a national survey estimated that Anti-HAV IgG prevalence is heterogenous among Brazilian capital cities [9,10]. This fact can also be verified by Hepatitis A occurrence. Health assistance services monitor newly diagnosed cases of acute hepatitis, which must report demographic, clinical, and laboratory information to compose the Viral Hepatitis Database of the National Reportable Disease Information System (SINAN) [11]. This information is the basis of temporal trend analysis and is crucial to understand Hepatitis A dynamics.

In the last decade, only a few studies analyzed Hepatitis A occurrence in Brazil over time [12,13,14]. As a rule, specific moments in time are defined a priori, and data are fitted to a linear model until and after these moments. This approach limits the analysis to previously known points where Hepatitis A occurrence changes. To overcome this limitation and improve the analysis, this study fitted Hepatitis A occurrence data to a joinpoint regression model to identify possible unknown points in time where the occurrence changed.

## 2. Methods

### 2.1. Data Source

Hepatitis A data extracted from the SINAN Viral Hepatitis database (SINAN Net—Version 5.0, Ministério da Saúde—Brasília, Brazil) from 2007 to 2018 were anonymized, reviewed, and depurated to constitute the study database. The variable indexing number (serving as an index variable for the database), dates of birth and notification, state of notification, gender, and Anti-HAV IgM (marked as reagent or not-reagent) were used for the analysis. Only individuals with reagent Anti-HAV were considered as a case of acute Hepatitis A (or acute HAV infection).

### 2.2. Statistical Analysis

Cases from 1 to 89 years old at notification were stratified by the following age groups: less than 5 years, 5 to 19, 20 to 39, and more than 40 years. Population age-stratified data from 2007 to 2018 of the Brazilian Institute of Geography and Statistics (Instituto Brasileiro de Geografia e Estatística—IBGE) [8] allowed the calculation of the national and regional notification rate of Hepatitis A, defined as cases per 100,000 inhabitants, by gender and age rates.

The software Joinpoint Regression Program 4.7.0.0 (NIH—National Cancer Institute—Maryland, USA) was used to fit the data to a joinpoint regression model with default parameters constant variance, and a maximum of 2 joinpoints. The model, selected by the Bayesian information criteria (BIC) [15], allowed the estimation of the annual percent change (APC) for time segments (e.g., from 2007 to 2011) and, then, the estimation of the average annual percent change (AAPC) for the full-time period (2007 to 2018), both expressed in percentages followed by its value with 95% CI [16]. The terms ‘increase’ and ‘decrease’ were used to describe temporal trends when APC or AAPC achieved statistical significance (*p* < 0.05). Otherwise, the term ‘stable’ was used. The software RStudio 2022.07.02 (Posit—Massachusetts, USA) was also used for data manipulation and analysis.

## 3. Results

### 3.1. Descriptive Analysis

In Brazil, 65,284 Hepatitis A cases were reported from 2007 to 2018. The Northeast region reported 18,732 (28.69%) cases while the North reported 18,430 (28.23%), the Southeast reported 14,073 (21.55%), the South reported 7909 (12.11%), and the Central-West reported 6140 (9.4%). The larger notification rate, cases per 100,000 inhabitants, was observed in the North from 2007 (13.91) to 2016 (1.64) when the Southeast surpassed all other regions until the last year of the study, as shown in Figure 1.

### 3.2. National Seasonality and Temporal Trends

National seasonality is shown in Figure 2. From 2007 to 2018, Hepatitis A occurrence achieved maximum values between January and June. An exception occurs in 2017 and 2018, where the peak occurs at the beginning of June and January, mainly in males. The Brazilian Joinpoint model, shown in Figure 3, estimated that the overall incidence of Hepatitis A in Brazil decreased from 2007 to 2018 (AAPC 17.5%, 95% CI −22.3% to −12.4%), without joinpoints.

### 3.3. Regional Temporal Trends

From 2007 to 2018, Hepatitis A occurrence in Brazilian regions decreased in the North (AAPC −24.5, −33.7 to −14.1), Northeast (AAPC −29.7, −35.5 to −23.5), Central-West (AAPC −25.1, −30.8 to −18.9), and South (AAPC −18.4, −23.2 to −13.4), while remaining stable in the Southeast (AAPC −2.6, −12.4 to 8.2). No joinpoints were identified in the Central-West region; one was identified in 2014 in the North, Northeast, and South, and one was identified in 2016 in the Southeast.

Regarding the North, Northeast, and South models, from 2007 to 2014, Hepatitis A occurrence remained stable in the North (APC −1.7, −15 to 13.7) while decreasing in the Northeast (APC −18.5, −26 to −10.3) and South (APC −27.7, −32.5 to −22.7). From 2014 to 2018, Hepatitis A decreased in the North (APC −52.4, −66.3 to −32.9), Northeast (APC −45.8, −56.8 to −32.0), and remained stable in the South (APC 0.8, −14.1 to 18.4). As for the Southeast model, Hepatitis A occurrence decreased (APC −15.7, −20.6 to −10.6) from 2007 to 2016 and remained stable (APC 86.6, −2.4 to 256.7) from 2016 to 2018. Table 1 summarizes these results.

Age group analysis of North, Northeast, Central-West, and South regions reveal that Hepatitis A occurrence decreased (AAPC) from 2007 to 2018 in all age groups below 40 years, while it increased or remained stable in southeast individuals above 19 years old. These results are presented in detail in Appendix A and illustrated in Figure 4.

## 4. Discussion

Temporal trend estimates of Hepatitis A occurrence reveal two patterns: first, for the North, Northeast, and Central West regions where the overall incidence decreases, mainly after 2014, and second, for the Southeast and South regions, where it decreases until 2016 and then increases mainly in males above 19 years old. The joinpoints identified in this study are the same defined a priori by other authors [12,13,14]. This result points out the strength of the methodology adopted for this work to identify changes in disease dynamics. Possible explanations for the encountered changes in Hepatitis A dynamics are the Brazilian socio-demographic heterogeneity, the improvements in national sanitation and the Brazilian vaccination program.

The North, Northeast, and Central-West regions have a higher social inequality index (Gini Index), a higher percentage of the population without access to sanitation, and less regular waste collection as compared to South and Southeast regions [17]. Therefore, it is expected that these regions have a higher incidence of Hepatitis A. Regarding South and Southeast regions, the lower Hepatitis A occurrence suggests that both regions are low-endemic regions for Hepatitis A. This hypothesis is strengthened by the Viral Hepatitis National Survey, already mentioned [9,10].

From 2007 to 2017, the Brazilian sanitation system was improved. Sewer extension increased from 184,000 km to 312,000 km and the population with access to treated sewers increased from 32.5% to 46% [17]. This directly impacts the HAV main route of transmission, therefore decreasing the occurrence of the disease, especially after 2014.

The most significant measure in favor of Hepatitis A control relates to the Brazilian vaccination program. A vaccine against Hepatitis A has been available in Brazil since the 1990s, but only in 2014 it was incorporated as a single dose into the National Immunization Program, encompassing infants from 15 to 24 months of age. In 2017, the program expanded to children less than five years old [18], and vaccine coverage achieved more than 90% of the target population [10,14]. The decrease in Hepatitis A incidence observed after 2014, mainly in individuals less than 20 years (see Appendix A for details), probably reflects a successful vaccination program.

Hepatitis A occurrence increased after 2016 in the South and Southeast regions, mainly in males between 19 and 39 years. This pattern resembles the epidemiologic pattern found in Europe, where most cases occurred in MSM [19,20,21]. Although the data available for this study do not allow the evaluation of this hypothesis, the results published by others related to Hepatitis A incidence from 2016 to 2018 point in this direction [22,23].

Before 2016, the main route of transmission probably was through contaminated water and food. Therefore, Hepatitis A occurred more frequently in the Brazilian summer and spring (end of December until the end of March) and affected both genders equally [24,25]. The change in seasonality pattern and the predominance amongst males (see Figure 2), that occurred after 2016, could be explained by sexual transmission among MSM. However, further studies regarding infection dynamics are necessary to explore this hypothesis.

Limitations of our work include the source of data and assumed hypothesis. Although revised, depurated, and consisted, it is not possible to guarantee the absence of duplicate cases in the database. Additionally, the “steady state” hypothesis concerning the process of notification in Brazil is not confirmed. However, unlike chronic hepatitis in the last decade, no new treatments against Hepatitis A were developed, nor were tracking campaigns realized. Therefore, if the notification process remained constant (“steady state”) from 2007 to 2018, it is possible to attribute reported case variation to true changes in disease dynamics. As for strong points, this study showed that the Joinpoint Model applied to Hepatitis A can identify moments in time where the Hepatitis A dynamics change. In contrast to other publications, these moments were not defined prior to the analysis. In addition, this study analyzed Hepatitis A occurrence in Brazil and provided a comparison between macro-regions, by age groups and gender using a methodology applied in other countries [26], allowing a proper comparison of world regions.

The results presented here reveal the success of vaccination programs and national sanitation improvement to control HAV transmission. Additionally, they suggest that Hepatitis A endemicity is heterogeneous within Brazilian regions. Therefore, regional differences must be accounted for in the strategic planning of control measures. Following this line of thought, after the identification of new cases in 2016, the Brazilian government expanded the Hepatitis A Vaccination Program. Since 2018, it included individuals who practice oral-anal sexual intercourse [27].

In conclusion, Joinpoint Models are an important tool for analyzing temporal trends. In addition, the models point to regional differences related to Hepatitis A endemicity and the existence of an adult population exposed to Hepatitis A that should be prioritized by control programs.

## Figures and Tables

**Figure 1 viruses-14-02737-f001:**
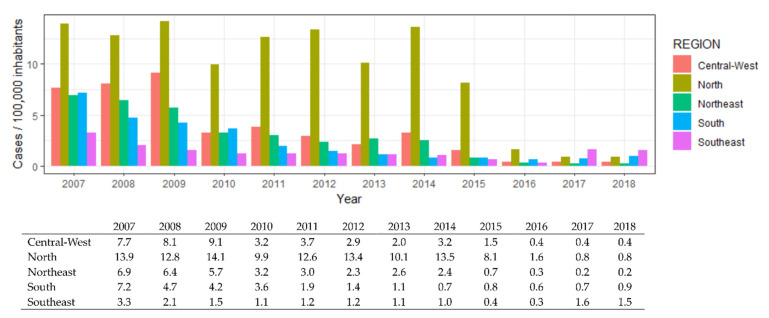
Hepatitis A cases per 100,000 inhabitants by year and Brazilian region with supporting data (cases/100,000 inhabitants by region and year).

**Figure 2 viruses-14-02737-f002:**
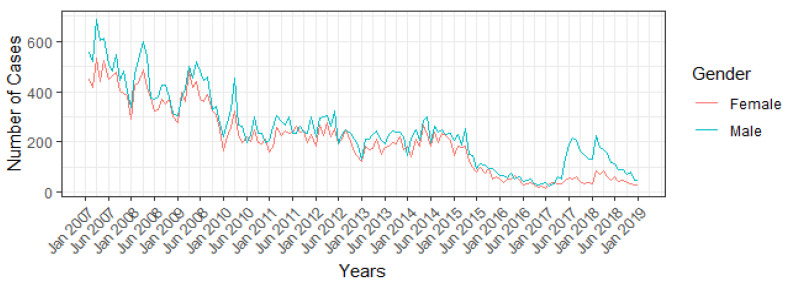
Hepatitis A reported cases in Brazil, by month and gender, from 2017 to 2018.

**Figure 3 viruses-14-02737-f003:**
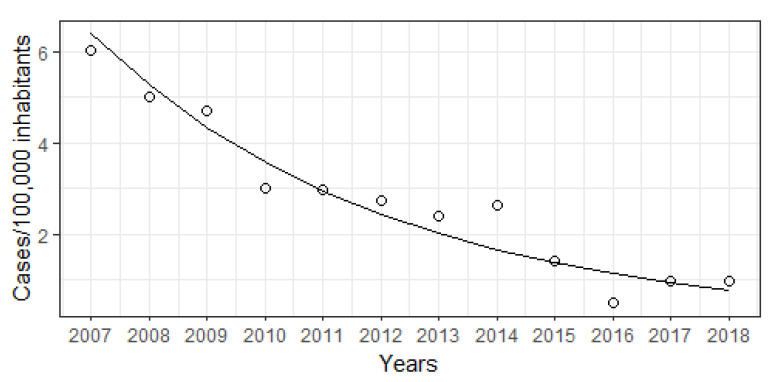
Hepatitis A observed (points) and modeled (line) occurrence in Brazil from 2007 to 2018.

**Figure 4 viruses-14-02737-f004:**
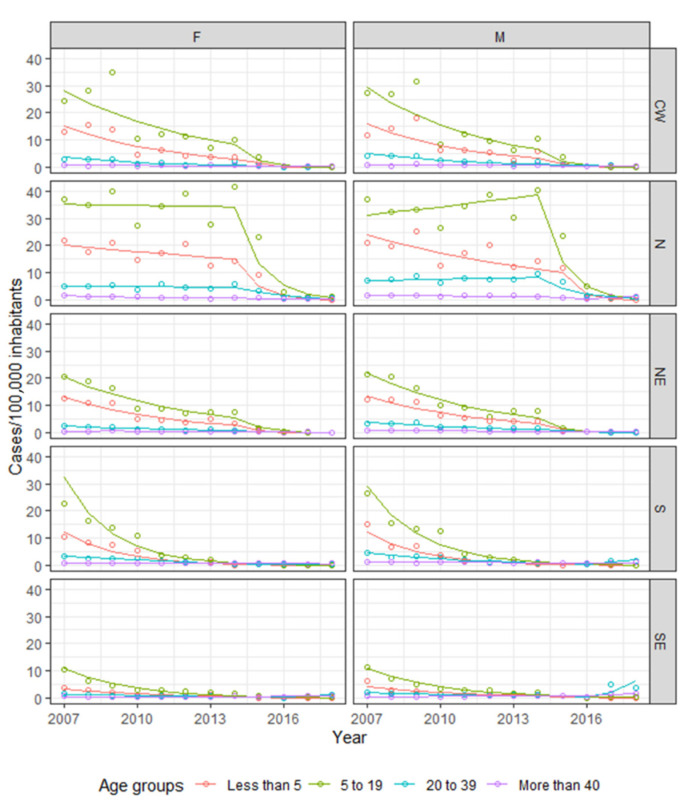
Hepatitis A observed (points) and modeled (lines) occurrence, in cases per 100.000 inhabitants, by gender and Brazilian region, from 2007 to 2018.

**Table 1 viruses-14-02737-t001:** AAPC and APC values with 95% CI of Brazil and its regions. In bold are trends that achieved statistical significance (*p* < 0.05).

Region	AAPC (%, with 95% CI)	APC (%, with 95% CI)
Brazil	**17.5, −22.3 to −12.4**	-
North	**−24.5, −33.7 to −14.1**	2007 to 2014: −1.7, −15 to 13.7 **2014 to 2018: −52.4, −66.3 to −32.9**
Northeast	**−29.7, −35.5 to −23.5**	**2007 to 2014: −18.5, −26 to −10.3** **2014 to 2018: −45.8, −56.8 to −32.0**
Central-West	**−25.1, −30.8 to −18.9**	-
Southeast	**−18.4, −23.2 to −13.4**	**2007 to 2016: −15.7, −20.6 to −10.6** 2016 to 2018: 86.6, −2.4 to 256.7
South	−2.6, −12.4 to 8.2	**2007 to 2014: −32.5 to −22.7** 2014 to 2018: 0.8, −14.1 to 18.4

## Data Availability

The datasets generated and/or analyzed during the current study are available in the SINAN—Sistema Nacional de Agravos de Notificação repository, http://portalsinan.saude.gov.br/ (accessed on 6 November 2020).

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
