# Peer review of "Temporal Trends of Acute Hepatitis A in Brazil and Its Regions"

_viruses, 2022, doi:10.3390/v14122737_

Round 1

Reviewer 1 Report

This study fitted hepatitis A occurrence data to a Joinpoint regression model to identify possible unknown points in time when the occurrence changed.

Findings provide important evidence overcoming the previous limitation and improving the analysis.

The results presented reveal the success of vaccination and national sanitation in the control of HAV transmission, suggesting that Hepatitis A endemicity is heterogeneous within Brazilian regions.

The discussion is enough consistent with the evidence andarguments presented.

The references result appropriate.

Author Response

We appreciate the comments made by the reviewer. Minor changes in the text were made to correct grammar and spelling issues, and improve the reading.  

Reviewer 2 Report

Using the Joinpoint regression analysis to identify trends in the occurrence of hepatitis A is a great idea. The manuscript needs some improvements. please see suggestions below:

Line 32: HAV abbreviation not well defined in text

Lines 32, 33: Consider revising the sentence. perhaps break it into 2: 1 about the disease of hepatitis A and 1 about the hepatitis A virus (HAV) transmission

Lines 34 - 37: The 'WHO position paper on hepatitis A vaccines - October 2022' include updated references for the highlighted sentences. ..."In 2019, Global burden of disease (GBD) data20 estimated 159 million acute HAV infections, resulting in 39 000 deaths and 2.3 million disability-adjusted life years.." (extract from the paper)

Lines 63-65: Any reasons why this aim is different that the preprint available online?

Lines 68-77: the 2 sections can be combined into one, i.e. 2.1. More details should be given for the case definition as per Bierrenbach et al, 2021 (https://doi.org/10.3390/vaccines9040407)

Line 86: confirm if SIC or BIC was used to estimate statistical model to be used. definition does not correlate with abbreviation.  

Lines 91-92: this should have been close to the beginning of the paragraph. the Jointpoint Regression software is used to compute APC and AAPC. the text should state whether default parameters were used or not. If not, details should be provided 

Lines 98-99: Provide supporting data for Fig 1 for each region and for each year or add values to each bar on the graph

Lines 104 -106: the seasonality mentioned in the text is not visible on figure 2, which shows reported cases per year NOT per month. Also seasonality is discussed further in the manuscript. 

Line 104: shown not show. consider verifying language for the whole manuscript

Line 113: supplementary table gives AAPC and APC by age group and by gender. It would be nice to have the data supporting Fig 3, i.e. overall APC and AAPC by age group or by region and show it on the fig

Lines 115 - 126: please summarise overall APC and AAPC data per region in a table or fig (as previously mentioned) in the manuscript; for better understanding.

Lines 128 - 131: provide more details on observations from fig4.  Also the trend of hepatitis A occurrence in individuals <20 yrs in the North (N) region has a different shape compare to the other regions. Please explain

Figure 4: CO is used for Central West instead of CW

Line 156: is it before or after?

Line 165 - 166: please point to supporting data in supplementary table, because Figure 4 only show light increase in 20-39 male in Southeast. If your results do not show it, shouldn't that be a limitation instead

Author Response

We appreciate the comments. Please find the author's answer in the attached file.

Round 2

Reviewer 2 Report

Thank you and well done. Your revised manuscript reads very well. Please see minor suggested revisions below:

Line 101: should be "...RStudio 2022.07.02 was also used ..." and not were

Line 141: should be "Table 1 summarizes.."  

Line 144: Title for table 1 should be above the table and not below

Author Response

We thanks the revisor for the comments. The suggested corrections were made.